# Behavioral and Physiological Reactions to a Sudden Novel Object in the Weanling Horse: Quantitative Phenotypes for Future GWAS

**DOI:** 10.3390/genes14030593

**Published:** 2023-02-26

**Authors:** Barclay B. Powell, Kelsey C. Horvath, Tyeler L. Gilliam, Kimberly T. Sibille, Andreas Keil, Emily K. Miller-Cushon, Carissa L. Wickens, Samantha A. Brooks

**Affiliations:** 1Department of Animal Sciences, University of Florida, Gainesville, FL 32611, USA; 2Departments of Physical Medicine & Rehabilitation, College of Medicine, University of Florida, Gainesville, FL 32611, USA; 3Department of Psychology and Center for the Study of Emotion & Attention, University of Florida, Gainesville, FL 32608, USA; 4UF Genetics Institute, University of Florida, Gainesville, FL 32610, USA

**Keywords:** heritability, Quarter Horse, heart rate, fear, startle, spook, equine, genetics

## Abstract

The startle response can be defined as a reflexive reaction to the sudden appearance of a novel stimulus that influences the survival and resilience of animals. In domesticated species, the behavioral component of the startle response can, in some cases, cause serious injury to the animal or human handlers if inappropriately expressed. Here, we describe a longitudinal study in a population of stock-type horses that quantified behavioral startle responses elicited by the presentation of a sudden novel object (rapidly opening umbrella). The study was performed in weanling foals across four consecutive years (*n* = 74, mean age = 256 days). Behavioral assays for the startle response phenotype focused on six behavioral variables: latency to return to the feed pan (seconds), maximum distance fled (meters), proportion of time spent walking or trotting (seconds), and how long a horse spent standing facing away from or toward the novel object. We observed behavioral startle response variables in relation to cardiac response, age, and sex for each individual. Each horse’s cardiac startle response pattern was determined and categorized into heart rate response cluster groups identified as accelerators and decelerators. Using principal component analysis (PCA) with a factor rotation, we identified “startle response” phenotypes that summarize the behavioral and physiological variables. The largest component of variation, Factor 1, comprised 32.5% of the behavioral variable with a positive correlation with latency and distance, and was not influenced by sex or age. Factor 2 comprised 23.2% of the variation, and was positively correlated with activity level performed such as proportion of time spent walking and/or trotting. Horses with the accelerator type cardiac response had significantly higher Factor 1 scores than decelerators but did not differ in Factor 2. Future work includes expanding our sample size to conduct a genome-wide association study (GWAS) to identify novel genetic loci influencing behavioral startle reactions using recorded behavioral and physiological phenotypes.

## 1. Introduction

The sudden response to a novel stimulus, known as the “startle response”, plays a key role in animal survival, as the ability to identify and escape danger draws the line between life and death. Startle-related behavioral traits studied in laboratory species, especially the murine model, reveal the impact of genetic polymorphism on highly conserved mammalian pathways contributing to variation in the startle response [1]. To better understand fear in mammalian species, startle tests expose an animal to a sudden visual and/or auditory stimuli to elicit behavioral and physiological responses [1,2,3]. Common physiological response measures include cardiac variations (heart rate changes) and locomotor reactions. Human safety is imperative when working closely with horses, and a horse with a higher propensity to startle with an escalated behavioral and/or locomotor response increases the risk of injury to both horse and handler.

The acute startle reaction involves responses on the behavioral, physiological, and cognitive level that are essential to survival in a threatening situation. The mammalian defensive response to a strong acute noxious stimulus has been studied for decades. Behavioral experiments and measures of peripheral physiology have led to the proposed defense cascade model [4,5]. A higher mean startle heart rate in untrained horses was found to be associated with a strong sympathetic nervous system response (fight or flight reaction) displayed during a novel object test. This increase in heart rate may occur independent of an increase in motor activity. Trained horses display significantly lower mean heart rate responses, likely due to improved vagal antagonism in response to learning [6].

This model suggests that the mammalian defensive response is a temporal sequence of events, ranging from the initial detection of a threat cue (orienting) to a circa-strike phase, which involves overt fight or flight. In this latter phase, behavioral and physiological data illustrate a massive mobilization for action, which often includes pronounced heart rate increase, locomotion, vocalization, and the subjective report of fear and panic [7]. Specifically, heart rate acceleration in the first 5–10 s after the onset of a full-cascade defensive episode (including detection, orienting, mobilization, and action) has been regarded as a metric of the intensity of the defensive response. As a consequence, a body of research has examined the extent to which heart rate acceleration during cardiac defense may serve as a robust index of interindividual differences in anxiety, stress reactivity, or related constructs [8,9]. In the laboratory, states of acute fear are often elicited by a disruptive noxious auditory stimulus such as a gunshot or other noise, unexpected by the participants. This procedure typically results in a complex sequence of changes in HR, with both accelerative and decelerative components that appear in an alternating fashion during the 80 s after stimulus onset [10,11].

Owners report fear-related behaviors as the single most common cause of human injuries from horse-related activities, so they are a trait critical to the sustainability of the horse industry [12]. The knowledge gap in the genetic basis of horse behavior is a barrier to the use of genomics tools for improved selection of behavioral traits, such as those related to fear. Genetic studies of behavioral traits are hindered by challenges in collecting quantitative and reliable phenotypes. To establish a quantitative and comprehensive phenotyping method for the startle response in horses, we adapted a behavioral assay intended to characterize behavioral and physiological indicators of startle, and then condensed these variables into a single ‘startle’ phenotype amenable to future genetic studies. Establishing an objective quantitative phenotype makes behavioral traits utilizable for genetic mapping. New genomic selection tools will allow for better herd management and improved sustainability of equine use for work, competition, and recreational activities.

## 2. Materials and Methods

### 2.1. Animals and Management

All horses were part of the University of Florida Equine Program, where they were bred, raised, and maintained under uniform management conditions and training schedules. The study cohort consisted of four consecutive foal crops (2014 to 2017, 15–22 foals per year) totaling 74 stock-type horses registered in the American Paint Horse Association, American Quarter Horse Association, and/or Appaloosa Horse Club (APHA/AQHA). At the time of testing, horses were between 188 and 315 days of age, and included 33 males and 41 females (see Appendix A). All foals were born in the spring, housed in groups of six to ten mare and foal pairs with close foaling dates in a 15-to-20-acre field. Foals were gradually weaned at approximately five months of age and kept with peers to minimize stress. Foals were transported to the University of Florida Horse Teaching Unit in their groups for the fall semester, where they receive introductory handling and training (e.g., haltering, leading, and basic ground handling) through an undergraduate weanling practicum course. Weanling horses in the practicum course are trained during approximately one-hour sessions, three days per week. All procedures were approved by the University of Florida IACUC (protocol #201509160).

### 2.2. Testing Environment

We adapted a novel object test previously described by Lansade et al. [2]. Foals were acclimated to the testing environment through a protocol implemented prior to behavioral testing. The testing area consist of a circular pen with a height of 2 m and diameter of 13 m with opaque mesh sides blocking visual distractions outside the testing area (Figure 1). A feed pan containing a food reward (grain mix), placed on the ground one meter from the fence, encouraged the horse to approach the novel object. The novel object, a brightly colored polka dot pattern umbrella, was placed through a gap in the side of the round pen approximately one meter off the ground above the feed pan. A high-definition digital camera (GoPro Hero3+ Silver, 720 p 60 fps, Go-Pro Inc., San Mateo, CA, USA) mounted on a ten-foot-tall PVC pipe was positioned above the test area to record all behaviors. Concentric circles (Figure 1), painted on the ground at 1 m intervals from the feed pan, aided in measurements of distance traveled and locomotor behaviors through video recording. A painted rectangle placed straight ahead when entering the pen, eight to ten circles from the feed pan, (Figure 1) denotes the location where the horses were released at the beginning of each test by their handler. 

### 2.3. Acclimatization

Each horse underwent a set of uniform acclimation and testing procedures in preparation for and throughout the testing period to ensure animal welfare and safety. The protocol allowed horses to acclimate to the testing area and procedure by habituating them to the pen, visual barrier, painted ground lines, stationary umbrella, and feed pan. Each horse was allowed to explore the pen unaccompanied by the handler for a period of five minutes to acclimate them to isolation from peers.

In the four to six weeks leading up to test day, students acclimated their horses to the test area in three stages. Horses were first led into the round pen and shown the location of the feed pan so they understood that a food reward was present in the pan during stage one. The second stage included the addition of the novel umbrella hanging closed and motionless above the feed pan. The third stage included the addition of the painted ground lines and required horses to be released in the center of the round pen, successfully seeking the food reward without assistance from their handler. This third stage was repeated three times to assure acclimation. Additionally, during their regular multi-week training sessions, horses were fitted with a girth strap at least three times prior to testing to allow habituation to barrel pressure from the heart rate monitor belt. 

### 2.4. Test Procedure 

On test day, each horse was fitted with a wireless heart rate monitor (Wear Link W.I.N.D. transmitter, Polar Electro, Inc., Woodbury, NY, USA) attached to an elastic girth strap for continuous recording of heart rate data. Heart rate data collections were initiated five minutes prior to entering the testing environment to capture baseline HR and continued until completion of the testing and removal of the horse from the testing area. Testing began when the horse was brought into the round pen and positioned facing the feed pan and novel umbrella in the start box by their assigned student handler. Then, the lead rope was removed, and the student exited the pen. After the horse approached the feed pan and performed three consecutive seconds of eating with their head in the pan, the horse was re-caught by the handler and led outside of the pen. The first trial run confirmed that on test day, each horse was successfully acclimated to the testing environment.

Immediately following the trial run, the horse was returned to the pen for the “test run.” After the animal ate for three consecutive seconds from the feed pan, the umbrella was abruptly opened. The horse had space to move freely throughout the pen in response to the sudden stimulus and then to re-approach the pan at will. Once the horse returned to eating from the feed pan for three consecutive seconds, the test ended, and the student handler retrieved the horse and led them away from the test environment. In the case that a horse did not return to the feed pan, the test concluded after five minutes.

### 2.5. Heart Rate Data Collection

Heart rate data were collected using the Polar heart rate monitor and Polar Protrainer Equine Edition software (Polar RS800CX, Polar Electro Inc.). Continuous HR and R–R interval recordings were performed using a Wear Link W.I.N.D. transmitter (Polar Electro, USA). Sensors (Wear Link) were attached to an elastic girth fitted around the barrel of the horse behind the elbow and tightened to bring the two electrode sensor pads into contact with the horse’s body. Baseline heart rates for each horse were collected for five minutes prior to approaching and entering the experimental area. Heart rates (HR) were continually recorded throughout the trial and test run until the handler removed the horse from the experimental arena after ending the test. 

### 2.6. Behavioral Recording 

Video recordings of each test session (GoPro Hero3+ Silver, 720 p 60 fps, GoPro, Inc.) were taken from the perspective of the top of the side wall, 90 degrees from the feed pan, providing a clear view of the entire experimental environment. The trial and test run for each horse were recorded in one continuous video session including the horse exiting the arena after the trial run and re-entering for the test run until the horse returned to the feed pan or failed to return within five minutes of the sudden novel object. One continuous video was recorded while each horse was wearing the heart rate monitor as a secondary verification method for interactions between the test environment, HR, and behavior. We characterized exhibited behaviors from these videos using Behavioral Observation Research Interactive Software [13] according to the ethogram described in Table 1. All measurements began at the time the umbrella was rapidly opened and ended when the handler returned to collect the horse from the testing area. Behavioral coding from video recordings was performed by two trained observers, with inter-observer reliability calculated for all behaviors observed, with Pearson correlation coefficients > 98%. Latency to return to the feed pan was measured during testing using live observation by a single observer stationed outside the testing area with a line of sight to the feed pan. 

### 2.7. Statistical Analysis

#### 2.7.1. Behavior Observations

Behavioral data analyses were performed using JMP Pro 16 (SAS Institute Inc., Cary, NC, USA). Duration-based variables (walk, trot, time spent facing toward or away from the umbrella) were converted into proportions by dividing the variables by the adjusted latency (total time spent in the test) to account for proportional time differences spent in the pen. Latency and flight distance values were adjusted to remove zero values to account for later transformation. Behavioral variables for time spent walking and trotting, standing facing toward and away from the umbrella, flight distance, and latency did not fit a normal distribution and were, therefore, normalized using a BoxCox transformation. A Shapiro–Wilk test showed that the proportion of walking, standing facing toward the umbrella, and standing facing away from the umbrella did not follow a normal distribution after transformation. A principal component analysis (PCA) was performed on the six behaviors, and a Varimax factor rotation for orthogonal transformation was used to correct for skewing in behavioral responses. Following rotation, we retained Factor 1 and Factor 2 to use as phenotype scores.

#### 2.7.2. Heart Rate Analysis

The electrocardiogram (ECG) time series were stored and converted online to time series of inter-beat intervals (IBIs), reflecting the interval between R-peaks in the ECG, defined by the Polar Software. The IBI time series was stored and used in subsequent analyses. Using Polar ProTrainer Equine Edition software (Polar RS800CX, Polar Electro Inc.), continuous heart rate recordings were evaluated to determine the heart rate changes occurring at the time the umbrella was opened during the test run. The validity of the IBI data during this segment was further examined by visual inspection. By aligning the Polar heart rate file to the recorded video, in addition to the noted startle time using a stopwatch, we identified the heart rate preceding and following the initial startle event the moment the umbrella was popped.

Epochs of 14 s were obtained from the continuously recorded HR data, including 2 s of baseline (pre-startle segment, −1 s), and 12 s after the event, using the startle event as a temporal reference point (0 s). IBIs that reflected missed R-waves or double R-waves (i.e., IBIs below 300 ms and above 1800 ms) were corrected by linear interpolation. IBIs were then converted to HR change values in beats per minute (bpm) values for each 1 s bin, as proposed by Graham [8]. The resulting waveforms represent how the HR changed relative to the pre-startle baseline, in bpm as a function of time, measured in 1-s bins.

Then, HR change data were evaluated in two steps. First the overall biphasic pattern (initial deceleration, subsequent acceleration) was quantified by scoring the mean HR change for each horse in two individual time segments: 2–5 s (initial deceleration), and 5–11 s after startle onset (subsequent HR acceleration). Second, following standard procedures for HR change studies in humans interested in inter-individual differences, the HR change waveform for each horse was submitted to a k-means cluster analysis, set to yield 2 clusters. This approach allowed us to categorize horses into two heart rate clusters characterizing the tendency to show behaviors consistent with freezing (sustained HR deceleration) versus flight reactions (deceleration followed by acceleration).

#### 2.7.3. Data Comparisons

Heart rate clusters and ethogram factor scores of behavioral variables were performed in JMP Pro 16 (SAS Institute Inc., Cary, NC, USA). Factor 1 and Factor 2 (see Section 2.7.1) were compared with the resulting heart rate clusters (accelerator or decelerator) using a one-way ANOVA. Factor 1 displayed statistical significance (*p* > 0.0003) with heart rate clusters, but no significance was found for Factor 2 (*p* > 0.5155) and heart rate clusters. Additionally, age and sex measures were compared to both ethogram factor scores using ANOVA with no significance for startle behaviors.

## 3. Results

Young horses exhibited diverse behaviors in response to the introduction of a novel object (Table 2). For example, the time to return to the feed pan post startle (latency) occurred in as short a time as two seconds, while some horses did not return within the time allowed (max experimental time of 300 s).

Young horses also demonstrated remarkable diversity in startle response as measured by the flight distance, which ranged from just 0.25 m to the furthest they could flee within the margins of the test pen, 12 m. The proportion of time spent walking and trotting differed across the sample population, with a maximum observed walking time of 224 out of 300 s in one horse. Young horses spent comparatively less time trotting than walking post startle (trotting a maximum of 80 s).

Following transformation, behavioral variables were combined into a single score using principal component analysis followed by factor rotation, yielding two summary “startle” scores. The first (Factor 1) component correlated most strongly to the latency, while the second component was driven largely by the flight distance. Factor 1 and Factor 2 captured 32.5% and 23.2% of the total variation, respectively (Figure 2).

Horses with high Factor 1 scores took longer to return to the feed pan (latency), fled a further distance from the novel stimulus, and faced away from the novel object while standing still, suggesting fearful behavior. Horses with low Factor 1 scores had lower latency times and smaller flight distances. In contrast, high Factor 2 scores describe horses who spent a higher proportion of time walking and trotting during the test. Thus, horses with both low Factor 1 and Factor 2 scores produced little response to the novel object, whereas horses with high Factor 1 and Factor 2 scores had both active and lengthy responses.

The age of the young horses varied from 188 to 315 days at the time of testing. Neither horse sex nor age affect the behavior factor scores (ANOVA *p* > 0.05). 

Continuous monitoring of heart rate during startle revealed two clusters of variation among the tested horses (accelerator and decelerator, Figure 3A), and these clusters significantly predicted the behavioral reactions captured by Factor 1 (*p* = 0.0003, ANOVA test) but not Factor 2 (*p* = 0.5155) (Figure 3B). Horses with accelerator-type cardiac responses exhibited higher Factor 1-type behavioral responses. For example, horses with an accelerator-type heart rate response tended to have increased startle reaction phenotype variables including longer latencies and greater flight distances. Decelerators had a lower mean Factor 1 score, describing startle responses with shorter latency time and smaller flight distances. Heart rate cluster did not affect Factor 2 scores, as Factor 2 captured aspects of continual locomotor activity in the behavioral response to a novel object.

## 4. Discussion

Investigating startle behaviors with concurrent physiological responses enables the development of a quantitative behavioral and physiological phenotype important for investigating genetic associations in the horse. We observed a significant effect of the immediate cardiac response on subsequent fearful behaviors summarized in Factor 1, but not the locomotor activity behaviors described by Factor 2. Horses with high Factor 1 scores tended to have an accelerated heart rate pattern in the 12 s following the startle event, whereas horses with low Factor 1 scores tended to have a decelerated heart rate after startle. Decelerator startle heart rate patterns characterized “freeze-type” behaviors in response to startle, for example, horses that did not flee as far from the novel object and had shorter latencies to return to the feed pan. Work conducted by Visser et al. [6,14] supports that heart rate (HR) and heart rate variability (HRV) are a reliable way to quantitatively assess temperament traits. Lansade et al. [2] later concluded that a fearfulness level could also be determined for individuals by observing specific behavioral reactions. Leiner and Fendt [15] combined the two approaches to investigate the interaction of HR and fear behavior. Fear behavior was positively correlated with an increase in mean heart rate (*p* < 0.0001) and a decrease in heart rate variability (HRV), which suggests a shift towards a sympathetic neural pathway when exposed to novel stimulus challenges [15]. The result of our study agrees with Leiner and Fendt’s findings, allowing for the development of a quality combined behavioral and physiological phenotype in our population amenable for future genetic study.

Visser et. al. found significant differences in mean startle HR between trained and untrained horses, highlighting the importance of environmental influence, experience, and management conditions [6]. These effects can be minimized through consistent management practices, controlled testing environments, and age specific testing. Heart rate recording enabled us to observe startle reactions more accurately in our population. In particular, we focused on the cardiac sympathetic modulation displayed at the moment of startle, thus minimizing the influence of post-startle physical exertion (i.e., locomotion behaviors, vocalization, etc.). Studies in rats have also documented startle behavior associated with significant variation in subsequent cardiovascular responses between normotensive inbred rat strains, suggesting genetic influence on cardiovascular responses [16,17]. Our results found a significant association between horses demonstrating accelerator startle heart rates and “fearful” behavior reactions in our population but did not find additional significant associations between the decelerator cardiovascular reactions and associated behavioral responses performed. If these pathways and systems are influenced by genetics, the significant association between “fear” behaviors and accelerator heart rates has the potential to uncover genetic ties to startle reactions in the horse, beneficial to the development of a better understanding of behavior genetics.

Horse owners, breeders, and handlers may benefit from genetic tools for the prediction of startle reactions, improving the ability of breeding programs to select for either more or less reactive horses to meet their breed or discipline goals. Genome-assisted selection for training programs would enable improved production and performance based on individual horses’ qualities, increasing value and economic benefits for owners and industry [12]. Understanding innate behavioral variations incorporating genetic approaches allows for the development of specialized training tools appropriate for the needs of each individual horse.

The development and analysis of the identified phenotypes support and strengthen our understanding of startle behavior and the concurrent physiological responses expressed during a startle event. The development of this new comprehensive phenotype model enables us to explore the interaction between the startle reflex and genetics. Blood and hair DNA samples were obtained from each horse for future genomic analysis of these new behavioral phenotypes. Resulting phenotypes discussed here will be used to perform a genetic analysis within our population to identify genetic controls responsible for startle behavior variations.

## 5. Conclusions

This study documents that variation in quantifiable behavioral responses is significantly affected by physiological reactions found in young horses after exposure to a sudden novel object. Heart rate patterns across the study population form two distinct groups expressing divergent physiological reactions. We have summarized a complex behavioral response into quantitative factor scores. These data provide phenotypes useable for future identification of genomic regions contributing to these traits in the horse. Focused evaluation of the heart rate cluster phenotypes with respect to behavioral reactions could help identify genes influencing startle reflexes to sudden novel stimuli to allow for selection of the desired variants. Equine breeding and use programs could benefit from additional tools in behavioral selection for safety and specific job selection to limit explosive or undesired startle reactions with greater potential for harm.

## Figures and Tables

**Figure 1 genes-14-00593-f001:**
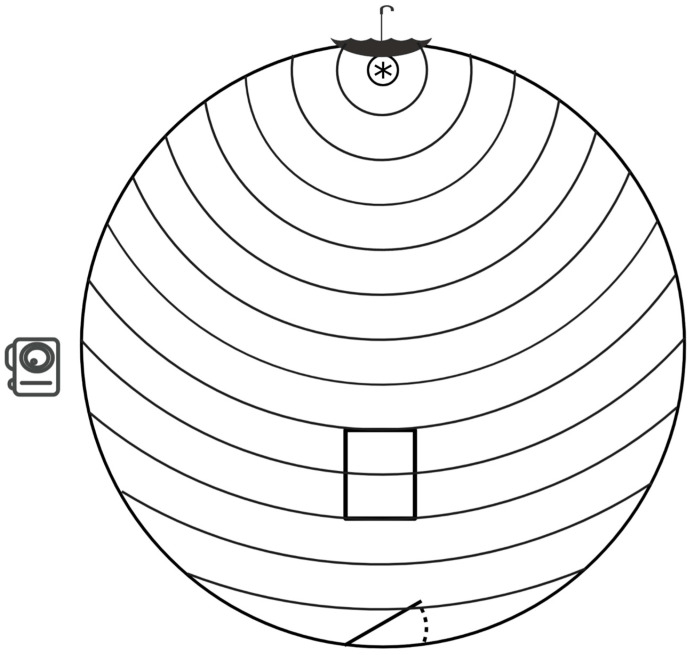
Diagram of test area layout, (not drawn to scale). Pan indicated by the asterisk in the pen
diagram. Umbrella symbolized by an umbrella symbol. GoPro camera symbolized by a camera image.
Line pointing upward to the right at the bottom of the figure with a dashed concave line symbolizes
the entry and exit gate. Rectangle in center represents release point from handler at the beginning
of test entry.

**Figure 2 genes-14-00593-f002:**
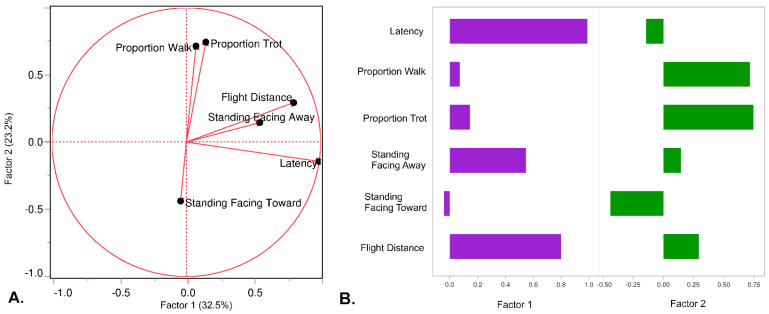
(**A**) The six behavioral variables detected using an ethogram were collapsed into two primary factors using PCA followed by rotation. (**B**) Two factors characterize behavioral traits such as “fearfulness” in Factor 1 and “activity” for Factor 2. Horses with a high Factor 1 score displayed a larger flight distance and took longer to return to the pan. In contrast, horses with a high Factor 2 score walked and trotted more frequently while in the pen.

**Figure 3 genes-14-00593-f003:**
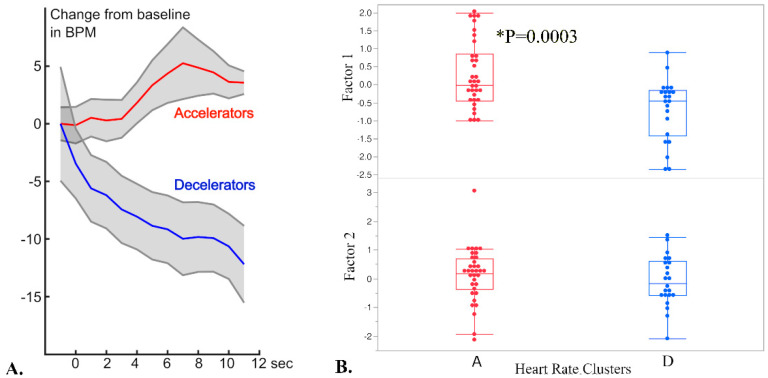
(**A**) Mean heart rate change time course relative to baseline, averaged across the members of the two clusters (red: accelerators; blue: decelerators; the grey shaded area defines the standard deviation). The time course is shown as relative change in BPM, for a time between −1 to +11 s, relative to the startle event. (**B**) Effect of HR clusters and Factor scores. Accelerator horse cardiac response types had significantly higher Factor 1 scores summarizing the latency, flight distance, and standing facing away from the umbrella behaviors.

**Table 1 genes-14-00593-t001:** Ethogram of behaviors observed through video recorded during the startle test.

Behavior	Description
Flight distance	Following umbrella opening, furthest distance from feed pan traveled during initial continuous motion; measured to nearest painted ground line from feed pan in meters
Walking	Moving at walking gait; frequency and duration
Trotting/cantering	Moving at either trotting or cantering gait; frequency and duration
Inactive attentive	Standing still with head and ears directed towards the umbrella; frequency and duration
Inactive not attentive	Standing still with head and ears faced away from the umbrella; frequency and duration
Latency to return to feed pan	Latency, following umbrella opening, to return to feed pan, lower head, and resume feeding. Assigned an upper limit of 300 s if the horse did not return.

**Table 2 genes-14-00593-t002:** Descriptive statistics of behavioral variables for startle response.

	Median	Minimum	Maximum	Mean	Std Dev
Flight Distance (m)	4.9	0.25	12	5.48	3.12
Walking Frequency	2	0	25	3.89	4.76
Trotting/Cantering Frequency	1	0	13	1.7	2.5
Defecation Frequency	0	0	3	0.162	0.524
Inactive attentive Frequency	1	0	12	1.68	2.13
Inactive not attentive Frequency	1	0	20	2.28	3.98
Return to feeder (yes/no)	1	0	1	0.946	0.228
Latency to return to feed pan (s)	20.75	2	300	55.89	82.83
Walking Duration (s)	11.016	0	224.273	27.96	42.57
Trotting Duration (s)	1.89	0	80.028	6.45	14.63
Inactive attentive Duration (s)	2.497	0	50.79	6.899	10.797
Inactive not attentive Duration (s)	1.38	0	192.535	13.467	34.077

## Data Availability

Data available upon request.

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
