# Peer review of "Behavioral and Physiological Reactions to a Sudden Novel Object in the Weanling Horse: Quantitative Phenotypes for Future GWAS"

_genes, 2023, doi:10.3390/genes14030593_

Round 1

Reviewer 1 Report

Review reports

A brief summary

This study idea is interesting as determining the behavioral and physiological phenotype of a specific behavioral reaction to negative stimulus as novel object that frighten animals till determine how to deal with it, is important in dealing with horses during training and work. However, the manuscript didn’t determine any genetic basis for such behavioral reaction but it is only a behavioral study that can be considered as stone for genetic work. It can be considered as an important behavioral study that consider the behavioral reflection to physiological reactions that happens in response to novel object. The manuscript needs to revised for clarification and improvement of materials, results, discussion and conclusion.

Specific comments 

Title:

-        Rewrite to behavioral and physiological reactions to a sudden novel object appearance in horse

Abstract:

-        Line 15-16: add (can be defined as) before (reflexive), remove (or), add (that) before (influences), write (of) not (in)

-        Line 24-25: add (components or variables) after (response), write (in relation to) not (was compared to), add (was determined) after (individual)

-        Line 27-28: write (identified) not (defined), write (response) not (behavior), add (that) before (summarizing), write (summarize) not (summarizing), remove (six), add (and physiological) before (variable)

-        Line 28-29: write (variables) not (variation), write (with) not (was), add (a) before (positive), write (positive) not (positively), (correlate) not (correlated)

-        Line 30-31: write (variable) not (variation), add (that) before (was), write (and/or) not (and), add (variables was) before (correlated)

-        Line 32: remove (associated with)

Introduction:

-        Line 39: write (that) not (frequently)

-        Line 45: remove (recorded), how locomotor behavior is a physiological response?

-        Line 50-54: too long, shorten, what you mean with connects human research with work in experimental animals?

-        Line 54-58: too long, unclear meaning, please clarify.

-        Line 76-77: not clear, clarify

-        Line 82: write (makes) not (creates), remove (a) before (behavioral)

Methods:

-        Line 90: mention number of horses per year

-        Line 92: is it the averaged age of horses per year or of the four years? The same for sex? Mention the age and sex number per year

-        Line 92-93: mention is it for each year?

-        Line 101: mention the name of researchers from which you adapted the testing procedure

-        Line 103: add (the testing area consists of ) before (a circular pan)

-        Line 106: the food pan is not a one meter distance as it is at opposite side which is 2 meters? Clarify

-        Line 142: third through fifth, is this the third stage?

-        Line 145: what is meant by regular multi week training sessions? How many times the horses were trained per week and for how long? Is the three stages is repeated every week or it is once at beginning?

-        Line 157: why there is a first trial on the test day although they receive an acclimatization period to test area?

-        Line 162: is the testing procedure was done on two days or two trials separately?

-        Line 164: you mentioned here that the umbrella is located one meter from the pan which is not mentioned above in the testing area but mentioned that it is one meter high above the pan, please clarify? Is this means that the first circle is for umbrella and second for feed pan?

-        Line 168: write (ended) not (concluded)

-        Line 170: is this means that the test lasts for five minutes so you continue for another two minutes if horse returned to feed pan after 3 minutes? what you mean by removed from test excluded or just get out of testing area?

-        Line 182: write (run) not (period), (ending) not (concluding)

-        Line 184: write (behavioral recording) not (behavior analysis)

-        Line 185: how many sessions of testing? How long is the session? Is it five minutes?

-        Line 190: write (ended) not (concluded)

-        Line 191: write (coding) not (observations), (was) not (were)

-        Line 193: write (with) not (and)

-        Line 196: what you meant by measured to nearest distance marker line (m)? is it the release point?

-        Line 198: behavioral observations not behavior analysis

-        Line 246: add (of behavioral variables) before (were)

-        Line 248: write (with) not (between)

Results:

-        Line 261: how many horses perform max walking or trotting duration?

-        Line 264: rewrite title of table to: descriptive statistics to behavioral variables of startle response

-        Line 267: add (component) before (correlated)

-        Line 268: add (component) after (second)

-        Line 269: clear factors on figure 2, is both factors include all behavioral variables?

-        Line 271-274: too long, shorten and clarify

-        Line 276: low or high proportion? Low or high factor 2? Not match line 277-279

-        Line 280-283: un clear

-        Line 282: write (affect) not (correlated to)

-        Line 291: where is the shaded error bars?

-        Line 294-296: write (effect) not (comparison)

-        Line 302: write (affect) not (correlate with)

-        N.B. where is the results of heart rate, figure 3A?

-        Line 305: what about decelerator ?

Discussion:

-        Line 309: add (and physiological) before (phenotype)

-        Line 310-311: write (effect of ) not (correlation between), remove (startle) before (response), write (on) not (and)

-        Line 316: not clear

-        Line 322: add (was found to be) before (positively)

-        Line 328: mention the study for who?

-        Line 330-335: too long, unclear

-        Line 336: what is the biological responses?

-        Line 337-341: redundant

-        Line 341-344: un related, remove

-        Line 348-360: reference?

Conclusion:

-        Line 363: write (affected by) not (correlates to)

-        Line 366-367: not clear

-        Line 370: write (behavioral) not (physiological)

Author Response

Our thanks to reviewer one for their thoughtful suggestions on how to improve our manuscript. We have strived to incorporate their comments wherever possible. We will note each change here with our response marked by the “>” symbol.

Reviewer 2 Report

The study is well designed and quite well described. Most of the comments are minor comments to improve the manuscript, however, the meaning of factor 1 and factor 2 must be better explained in the abstract and in the manuscript.

Title: Is it to a visual or auditory new stimuli? Is it in the horse or weanling foal?

Line 24: ‘physiological variables such as cardiac response, age, and sex’. Are those physiological variables?

Line 28:’ Factor 1, comprising 32.5% of the variation’ this is not clear for the reader please explain.

Line 29: ‘Factor 2, 29 comprising 23.2% of the variation’ this is not clear for the reader lease explain

General remark concerning the abstract: what is the sudden novel object? Cfr title

Line 94: what do you mean with slowly separated?

Line 101: why do you mention this reference? Describe how and what you test.

Line 127: ‘or others’ what do you mean with others? Things in the environment?

Line 215: which QRS detection algorithm?

Line 242: I wander if this is freezing or fast acclimatization?

Line 246: what are factor 1 and factor 2? Explain please.

Table 2: can you add units (meters, seconds ??)? Maximum flight distance is 13 in the text and 12 in the Table. Defecation is not mentioned in the text. There were 74 foals observed . 12 are inactive attentive, 20 inactive not attentive, 1 returned to feeder. It is not clear what the others are doing?

Line 348-350: this can be of value in some branches but in some disciplines there are other factors that are more important.

Line 350: did you collect blood/hair of these foals?

Author Response

Our thanks to reviewer two for their thoughtful suggestions on how to improve our manuscript. We have strived to incorporate their comments wherever possible. We will note each change here with our response marked by the “>” symbol.

Round 2

Reviewer 1 Report

A brief summary

The manuscript was greatly improved for clarifying the aim of the study and its future work basis. Here below fewer editing.

Specific comments 

Abstract:

-        Line 25: cardiac response, age and sex is not a behavioral variables so add (in relation to) before (cardiac response), remove (such as)

Introduction:

-        Line 58-59: remove (a), add (was found) before (associated), remove (is) before (associated)

-        Line 61: write (this) not (the)

Methods:

-        Line 100-104: different number between text and table for male and females, revise

-        Line 156: how many weeks?

-        Line 177: for how many times the procedure was repeated?

-        Line 206-209: why record the trial run although it was not used in measurement analysis?
